# Low Protein Diets and Plant-Based Low Protein Diets: Do They Meet Protein Requirements of Patients with Chronic Kidney Disease?

**DOI:** 10.3390/nu13010083

**Published:** 2020-12-29

**Authors:** Daniela Verzola, Daniela Picciotto, Michela Saio, Francesca Aimasso, Francesca Bruzzone, Samir Giuseppe Sukkar, Fabio Massarino, Pasquale Esposito, Francesca Viazzi, Giacomo Garibotto

**Affiliations:** 1Department of Internal Medicine, University of Genoa, 16132 Genoa, Italy; daverz@libero.it (D.V.); danipicciotto@me.com (D.P.); michela.saio@virgilio.it (M.S.); pasquale.esposito@unige.it (P.E.); francesca.viazzi@unige.it (F.V.); 2Clinica Nefrologica, Dialisi, Trapianto, IRCCS Ospedale Policlinico San Martino, 16142 Genoa, Italy; 3Clinical Nutrition Unit, IRCCS Ospedale Policlinico San Martino, 16142 Genoa, Italy; francesca.aimasso@hsanmartino.it (F.A.); francesca.bruzzone@hsanmartino.it (F.B.); samir.sukkar@hsanmartino.it (S.G.S.); fabio.massarino@hsanmartino.it (F.M.)

**Keywords:** CKD, low protein diets, plant-based low protein diets, amino acid

## Abstract

A low protein diet (LPD) has historically been used to delay uremic symptoms and decrease nitrogen (*N*)-derived catabolic products in patients with chronic kidney disease (CKD). In recent years it has become evident that nutritional intervention is a necessary approach to prevent wasting and reduce CKD complications and disease progression. While a 0.6 g/kg, high biological value protein-based LPD has been used for years, recent observational studies suggest that plant-derived LPDs are a better approach to nutritional treatment of CKD. However, plant proteins are less anabolic than animal proteins and amino acids contained in plant proteins may be in part oxidized; thus, they may not completely be used for protein synthesis. In this review, we evaluate the role of LPDs and plant-based LPDs on maintaining skeletal muscle mass in patients with CKD and examine different nutritional approaches for improving the anabolic properties of plant proteins when used in protein-restricted diets.

## 1. Introduction: Why We Need Proteins of a High Biological Value

Although ~10%–15% of the energy in the diet of the Western world is derived from protein, humans do not need protein as a fuel but to increase their body cell mass during growth and anabolism and to maintain it under steady-state conditions [1]. In 1951, Rose, Haines, Warner and Johnson [2] formulated their classification of the nutritional significance of the individual amino acids (AAs), which were derived from their studies on the maintenance of body nitrogen (N) equilibrium in young men. In their classification, lysine, leucine, isoleucine, valine, tryptophan, phenylalanine, threonine and methionine were defined as “essential,” whereas the remaining common AAs were defined “nonessential”AAs (NEAAs). Later [1], histidine was added to the list of essential AAs.

If the essential AAs are lacking, neither growth in infants nor positive N balance in adults can be maintained. The NEAAs contained in protein can be synthesized from other precursors, such as glucose or tricarboxylic acid intermediates (pyruvic acid, oxalacetic acid and α-ketoglutarate) or from other essential or NEAAs. 

Even if plants and bacteria are able to synthetize all the AAs contained in proteins, animals do not possess all the required enzymes. The synthesis of AAs in humans comprises several pathways in which AAs are built from other precursors. The carbon skeletons of AAs are derived from seven intermediates, which are found in the glycolytic, pentose phosphate and citrate cycle pathways [3].

Once ingested, proteins need to be hydrolyzed by proteases and peptidases to AAs, dipeptides or tripeptides in the lumen of the small intestine to offer their nutritional value. Thus, the protein nutritional value depends not only on its content, but also on digestibility and relative AA proportions. Some AA intermediates are toxic, as is seen in various genetic AA disorders. Early studies have shown that in humans fed a protein-adequate diet, an excess of a specific AA causes an increase in its oxidation, while the oxidation of other AAs is unchanged. Contrariwise, when the dietary intake of an essential AA is insufficient, the oxidation of other AAs increases progressively with the increase of protein intake (DPI) [1]. In a specific protein, the essential AA present in the lowest proportion relative to the essential AA pattern requirement is referred to as the “limiting” AA. Protein synthesis depends on the availability of this “limiting” essential AA, which also determines the rate of utilization of all other AAs ingested with food [1,4].

The capability of protein to meet individual essential AA requirements is a reflection of its quality [5]. Protein quality, its digestibility and bioavailability can be affected by several factors, including food processing, cooking and storage conditions [6,7]. From a nutritional perspective, there is no requirement for protein per se, but rather a requirement for the essential AAs that compose the same protein [1]. An example of different body needs regarding individual AAs is offered by their different release in plasma from the body sources in the post-absorptive, overnight fasted state. Plasma fluxes of some NEAAs, such as glycine and alanine, are very high if compared to their relative concentration in proteins. Therefore, these AAs are synthesized at rates in excess of their needs for protein synthesis. However, for other AAs, such as arginine and tyrosine, fluxes are close to their concentrations in proteins, indicating much lower synthesis rates [3,8,9,10]; therefore, their body production may be insufficient in conditions characterized by increased AA demand (such as sepsis, acute kidney injury etc.).

On the basis of recent research, the dietary significance of individual AAs is even more complex than previously theorized [1,9]. Some previously considered NEAA, such as arginine and tyrosine, can be produced by humans, but their synthesis is limited by a variety of factors, including the dietary supply of precursors and disease state. These AAs are, therefore, considered to be ‘conditionally essential’. Depending on the physiological and metabolic condition of the host, “conditionally essential” AA may include arginine, tyrosine, proline, glutamine, cysteine, glycine and taurine [1]. 

In addition, some ‘requirement’ for NEAAs may actually also exist [10]. International agencies set a safe protein intake at 0.8 g/kg body weight per day in adult man, whereas the sum of the recommended daily allowance (RDA) of the essential amino acids is ∼0.2 g/kg body weight per day. Therefore, the resulting NEAA requirements would be ∼0.6 g/kg body weight per day [10]. 

In humans, as much as 200–250 g of proteins is broken down over a day, and their constituent AAs are in large part re-used in protein synthesis [11]. Protein turnover needs a large reutilization of AAs, which are released by protein breakdown into protein synthesis (hence, a great “efficiency” of protein turnover): as much as 60 to 65% of AA deriving from muscle proteolysis is cycled back into muscle protein synthesis in the post-absorptive state. Under nonrestricted diets, only 12–15 g of N is eliminated into the urine daily, reflecting the net catabolism of approximately 75–95 g of protein [12]. These lost AAs need be replaced from the diet or synthesized in the body.

Besides its AA composition, the nutritional value of dietary proteins depends on several other factors. Biological value, net protein utilization, digestibility, chemical score and digestibility-corrected AA score are currently considered for the evaluation of quality of dietary protein [5,6,7,13]. Many of these factors are better for animal-based protein sources such as milk, eggs and meat than for plant-based protein sources [14]. Another measure which is used to evaluate the ability of dietary protein to meet the body’s AA requirements is the Protein Digestibility Corrected AA Score (PDCAAS] [13,15]. Better than the previously used PDCAAS, the nutritional quality of proteins is currently evaluated by the Digestible Indispensable AA Score (DIAAS] [15], where:

DIAAS% = 100 × ((mg of digestible dietary essential AA in 1 g of the dietary protein]/(mg of the same dietary essential AA in 1 g of the reference protein]]. 

DIAAS corrects some of the drawbacks of the PDCAAS method by considering the ileal digestibility of each AA preferably determined in humans or in the growing pig (as compared to the rat in the PDCAAS). According to the Food and Agricultural Organization of the United Nations (FAO) DIAAS report, proteins are classified after their DIAAS value: <75 (no quality claim), 75–99 (high-quality protein) and ≥100 (excellent quality protein). The FAO recommends that the general population be categorized according to three distinct age-related reference patterns: 0–6 months (infant), 0.5–3 years (children), and >3 years (rest of the population) [15].

Protein quality based on DIAAS differs not only between animal and plant proteins but also between plant protein sources. Recently, Herrerman et al. [16] calculated DIAAS for several animal and plant proteins (Figure 1).

Based on the 0.5- to 3-year-old scoring pattern, potato and most animal-derived proteins are close to the excellent protein quality category (DIAAS ≥ 100). However, whey and soy proteins are in the high-quality protein range (DIAAS = 75–99). Several plant-sourced proteins are in the no quality claim category (DIAAS < 75). The differences in the protein quality offers the opportunity to enhance their nutritional efficiency by combining them. Potato, soy and pea proteins seem the most important plant-based complementary vegetal protein sources to reach high quality mixtures in a plant-based diet [16]. 

However, AA deficiency may appear if only one of the essential AA is present in inadequate amounts in the diet. As an example, a deficiency in tryptophan and lysine can take place if the diet is based mainly on grains that contain relatively low amounts of these AAs [17,18,19]. 

Finally, the biological value of a protein can be influenced by disease [20]. 

## 2. Aging and Sarcopenia

The ability of a muscle to respond to anabolic stimuli such as food ingestion is impaired in the elderly [20,21,22]. This ‘anabolic resistance,’ which implies that more proteins are necessary to appropriately increase muscle protein synthesis as compared to young subjects, makes elderly subjects less able to maintain N balance during fasting and increases body cell mass after feeding. This ‘anabolic resistance’ to food is suggested to play a major role in sarcopenia [20,21], which may result in increased risk of falls, frailty, loss of independence and poor life quality [22]. Of note, sarcopenia has been recognized as a disease by the World Health Organization (WHO). According to all of these findings, it has been strongly suggested to optimize protein intake in the elderly, both by increasing protein quantity (∼1.2 g/kg body weight/day) and improving protein quality [23].

## 3. Anti-Anabolic and Catabolic Factors in CKD

Several authors have shown strong associations between muscle mass surrogates and survival in patients with CKD [24,25,26]. Inadequate nutritional intake is very common in CKD5-5d patients [24]. It is interesting that many patients develop progressive weight loss 3–6 months before their beginning of maintenance dialysis, suggesting that anorexia and/or accelerated catabolism are responsible for wasting. It is now clear that both increases in anorexigenic hormones [25] and activation of proinflammatory cytokines [26] are associated with anorexia. However, it is now clear that wasting is due to abnormalities that stimulate protein degradation [27] and/or decrease protein synthesis [28]. Insulin/IGF-I resistance (in CKD3-5), acidosis and inflammation (in CKD5d) affect different intracellular signals and pathways that promote protein degradation [27,28,29] (Figure 2), such as caspase-3 mediation of apoptosis and activation of the ubiquitin-proteasome system. A decrease in muscle protein synthesis is also a newly recognized mechanisms for wasting, especially in patients with inflammation. Recently, Zhang et al. [28] were able to demonstrate that CKD stimulates chromatin-modifying, nucleolar protein 66 (NO66), which suppresses both ribosomal DNA transcription and muscle protein synthesis via a demethylase mechanism.

The activation of catabolic pathways proceeds along with the progression of CKD. Anorexia, acidosis, altered intracellular insulin signaling and inflammation overlap with catabolic/anti-anabolic pathways already operating in ageing and in comorbid conditions, such as diabetes and sepsis, to orchestrate the protein-energy wasting syndrome [29].

## 4. AA Requirements in Clinically Stable Patients with CKD

Patients with CKD3-4 present several abnormalities in AA blood, tissue and inter-organ exchanges, findings which suggest that CKD is a disease-state in which individual AA requirements are different from normal. The metabolism of branched-chain AA (BCAA) (leucine and valine), arginine, tyrosine, tryptophan, cysteine and of several essential AAs are altered [30]. In addition, the concentration of NEAAs has been reported to be abnormally high, mainly after meat ingestion [30]. However, a few studies have specifically addressed individual AA requirements in CKD and the results are hindered by AA toxicity issues. A recent hypothesis is that processing of the ingested AA by gut microbiota may yield uremic toxins. As an example, following tryptophan ingestion, the gut may produce p-cresyl sulfate, indoxyl sulfate and indole-3 acetic acid [31]. In addition, uremia-specific alterations in muscle AA metabolism are likely involved in the acceleration of catabolic processes or in reduced anabolism. Despite a major role of the kidney on ketoisocaproate oxidation [32], low levels of ketoisocaproate, the leucine ketoanalogue, occur in plasma and in muscle cells of CKD patients [30]. These abnormalities arise largely because of reduced nutritional intake, hyperinsulinemia, urinary protein losses and increased muscle BCAA catabolism [30]. Overall, in CKD3-5, the abnormality in the pattern of blood AAs resembles that observed in malnutrition, and can be further aggravated by low nutrient intake [30].

The loss of metabolizing renal tissue can account for several of the AA abnormalities found in CKD. Circulating levels of some AAs, such as 3-methylhistidine and 1-methylhistidine, rise because of their reduced renal clearance. Moreover, in CKD patients, the kidney removal of citrulline, s-adenosyl-homocysteine (SAH) and glutamine is blunted and the synthesis of several AAs which are newly generated within the kidney, such as arginine, serine and tyrosine, is reduced [31,33,34,35].

The abnormal arginine supply to tissues in uremia may have negative effects, since arginine is the precursor for nitric oxide (NO). A diminished availability of NO is implicated in the genesis of hypertension, function and integrity of endothelial cells, progression of CKD, accelerated atherosclerosis and also depression and behavioral changes [35,36]. Moreover, since NO is a mediator for several hormones, including Insulin Growth Factor-I (IGF-I) and Growth Hormone (GH), a reduced NO availability could hinder hormone action. In addition, arginine serves as a methyl donor for regulation of many biological processes including transcription, and DNA damage repair. 

The issue of the requirements for tryptophan in CKD patients is also unsettled. Tryptophan is essential for protein synthesis in humans. While free tryptophan levels are low in CKD, downward metabolites of its catabolism accumulate with progressive renal disease, owing to increased production of kynurenine by the enzyme indoleamine dioxygenase-1 [37]. Low tryptophan levels may account for incident cardiovascular disease and decrease in melatonin synthesis [38,39].

Finally, patients with CKD show an abnormal pattern of sulfur blood AAs [30]. Free and protein-bound cysteine, s-Adenosyl-Homocysteine (SAH), homocysteine (Hcy) as well as cysteinsulphinic acid (CSA), are high in plasma of uremic patients, while methionine is normal and taurine is low [30]. Major alterations are a decrease of Hcy clearance by both the transsulfuration and remethylation pathways [33,34]. A possible feed-back inhibitory substrate for the transsulfuration pathway is sulfate, the end-product of methionine catabolism. The decrease in urinary sulfate excretion causes progressive sulfate accumulation in blood, which is responsible for part of the increase in the anion gap observed in patients with end-stage renal disease (ESRD). Recently, methionine requirements have been studied in children with CKD3-4 [40,41]. Children with CKD have a similar mean and population-safe sulfur AA requirements to that of healthy children, but higher minimum methionine requirement, in keeping with reduced remethylation rates. However, methionine supplementation might increase in blood SAH [42] and Hcy [42,43], which are associated with increased risk of cardiovascular disease (CVD), and have been the target of several secondary prevention trials of high-dose vitamins, including folic acid, cobalamin and pyridoxine [43,44]. A recent meta-analysis [43] indicated a 10% lower risk of stroke and a 4% lower risk of overall CVD with folic acid supplementation. A greater benefit of folic acid supplementation for CVD was observed among participants with lower plasma folate levels and in studies with larger decreases in homocysteine levels. 

## 5. Dietary Protein Requirements in Clinically Stable Patients with CKD

The dietary protein requirements of clinically stable, non-nephrotic, non-acidemic, non-inflamed adult CKD patients appear to be similar to those of normal healthy subjects [45,46]. However, the recommended DPI in maintenance dialysis patients is much greater, (∼1.2–1.3 g/kg/day) [45,46]. Since in non-dialyzed patients protein restriction has multiple advantageous effects, including the decrease the accumulation in blood and tissues of many toxic intermediary products, the “standard” recommended LPD provides 0.55–0.80 g protein/kg/day [45], with more than 50% of the protein of high biological value. Very low (0.3–0.4 g protein/kg/day) protein diets (VLPDs), supplemented with AAs and/or ketoacids (KA) (7–15 g mixture of KA or hydroxyacid analogs of five essential AAs plus tryptophan, histidine, threonine and lysine) are more commonly offered to patients at an advanced CKD4-5 state. A variant of this diet, a 0.7 g protein/kg/day vegan diet, also has been proposed [45,46]. This diet provides more vegetables and fruits, which may have other advantages for CKD patients [47,48]. 

A high energy intake is a necessary component of CKD nutritional treatment. The association between dietary energy and N balance has been studied in clinically stable CKD patients fed with different energy intakes, at the same low protein intake (0.55 to 0.60 g/kg/day) [49]. N balance was directly related to energy intake, with an intake providing approximately 35 kcal/kg/day more likely to maintain a neutral or positive N balance. Thus, an adequate energy supply is an important aspect of nutritional treatment in CKD [50].

## 6. Does Aging Increases Protein Requirements in Patients with CKD?

The issue of protein requirements in CKD is more complicated if we consider that people over the age 65 are expected to soon become the majority of those who will need renal replacement therapy. As compared to younger adults, elderly subjects are less hungry, consume smaller meals and have lower energy intakes [51]. Weight loss in elderly subjects is predictive of increased morbidity and mortality [52,53,54,55] and it is supposed to be due to microinflammation and excess cytokine production (“inflammageing”). In addition, in elderly subjects, a decreased sensitivity of insulin action is linked to frailty and cognitive impairment [56]. The traditional recommendation for protein intake of 0.8 g/kg/day for adults of all ages is currently debated for elderly subjects, considering that they need higher amounts of protein (1.0–1.2 g/kg) for preservation of lean body mass [21].

In a recent study, the administration of a diet providing 2 RDA for protein compared with the current guidelines was followed by increased lean body mass and leg strength in elderly men [55]. Therefore, restricting protein intake in elderly CKD subjects might cause or aggravate sarcopenia. In elderly CKD subjects, a recent guideline suggests a minimum protein intake of 0.8 g/kg when estimated glomerular filtration rate (eGFR) is less than 30 mL/min [56]. However, some observations suggest that individual protein requirements may be diminished in CKD3-4 as compared to normal subjects, in the absence of concurrent catabolic risk factor. As an example, a study on muscle protein metabolism including elderly CKD patients has shown that these patients are minimally catabolic [57]. Additionally, available clinical studies suggest that elderly subjects with CKD can successfully adapt to a 0.6 g/kg LPD and to a supplemented VLPD. As an example, the Diet or Dialysis in Elderly (DODE) trial, which included 112 elderly patients with CKD [58], observed very modest nutritional changes following VLPDs supplemented with AAs and KAs and high calorie diets, suggesting that supplemented VLPDs are nutritionally safe in patients compliant with these diets. However, a definite response can be offered only by studies having specific end points, such as lean body mass, strength and physical function.

## 7. LPD in Patients with CKD: Indications and Advantages

The elimination of N waste products deriving from protein and AA catabolism progressively decreases during the course of CKD, resulting in retention of “uremic toxins” (among which the most known are hydrogen ion, phenols, indoxyl compounds, guanidines and advanced glycation end products) [59,60,61]. Accumulation of uremic toxins contributes to the onset of anorexia and nausea with subsequent decrease in nutrient intake [47,60] and progressive depletion of protein and energy body stores [47]. Uremic toxins may also contribute to the onset of a micro-inflammatory state, more frequent at an advanced CKD5 stage [61]. 

The generation of many uremic toxins diminishes when protein intake is restricted. However, nutritional treatment of CKD is also based on careful attention to the intake of phosphate, sodium, protein quality and energy. Besides reducing the burden of uremic toxicity, other major targets are the reduction of progression of renal insufficiency and proteinuria, a better control of blood pressure, complete correction of metabolic acidosis and maintenance of CKD/MBD parameters into acceptable targets. Another major goal is the improvement of compliance to nutritional treatments. Potential uses and benefits of AA/KA supplemented VLPDs for far advanced CKD patients are reviewed elsewhere [61].

## 8. The Adaptation to an LPD in Patients with CKD 

Early studies have shown that young CKD subjects on a high energy intake can maintain a neutral or even slightly positive N balance with protein intakes as low as 0.55–0.6 g/kg [45,47,48,49]. Studies performed using stable isotope AA kinetics have shown that adaptation to dietary protein restriction is achieved through a decrease in the rate of AA flux and oxidation leading to more efficient use of dietary AA and reduced ureagenesis [60]. Of note, the concept of “adaptation” to low protein intakes is very far from the concept of “accommodation,” the latter term implying a decrease in protein synthesis, with development of wasting, when DPI becomes inadequate, i.e., beyond the limits of the adaptive mechanisms; an impaired ability to activate an adaptive response might impair N conservation when an LPD is prescribed [60,62,63].

Protein turnover needs large reutilization of AA released by protein breakdown. In CKD patients on a 1.1 g protein/kg/day, similarly to the normal condition, as much as 60 to 65% of AA deriving from muscle proteolysis is cycled back into muscle protein synthesis [60]. This percentage increases by 12% when patients ate a 0.55 g/kg LPD, and increases slightly with a VLPD supplemented with AA/KA [60]. Therefore, in the postabsorptive state, during the adaptation to an LPD or a supplemented VLPD, protein turnover is characterized less negative net balance, decrease in protein degradation and more efficient AA recycling. An increase in insulin sensitivity may account for the increase in efficiency in protein turnover [60]; in addition, muscle autophagy has been shown to be reduced when an LPD was supplemented with leucine and KAs, as compared with an LPD, in rodent CKD models [64,65]. 

Preservation of muscle protein synthesis is necessary for a successful adaptation to a low protein intake; however, the minimal protein intake to maintain skeletal muscle protein synthesis in healthy subjects is still unknown. In a recent study, a 10–12% decrease in whole body protein synthesis and degradation was observed during a 12-week adaptation to a low (0.4 g/kg) vs. high (2.4 g/kg) biological value protein intake [66]. However, the 0.4 g/kg protein diet did not lower fractional muscle protein synthesis, suggesting that muscle protein synthesis in some healthy young individual may be unexpectedly maintained at very low protein intakes. However, a 0.45 g/kg day protein intake is not sufficient to maintain protein synthesis and muscle mass in elderly subjects [67,68]. Early attempts with the use of high biological values protein intakes as low as 0.25 g/kg, even if supplemented with KAs, have shown the occurrence of wasting [69]. However, a recent Cochrane metanalysis [70] shows that, in young/middle aged CKD patients, LPDs (protein intake set at 0.55–0.6 g/kg) and supplemented VLPDs (0.3–0.4 g/kg), when properly implemented, do not engender protein-energy wasting.

## 9. The Several Advantages of Plant-Based LPD in Patients with CKD

In plant-based LPDs (a broad term which includes vegetarian, vegan, Mediterranean and flexitarian diets) [71] currently used in CKD, DPI varies from 0.6 to 0.8 g/kg/day with at least 50% plant-based foods, usually unrefined and unprocessed foods. In addition, plant-based LPD contain low sodium (<3 g/day), high dietary fiber (25–30 g/day) and adequate dietary energy intake (30–35 cal/kg/day) [57] and 30 kcal/kg/day for those aged ≥ 60 year [71]. Plant-based LPD for CKD patients are encountering increasing success for many reasons. Several studies support the concept that in healthy subjects red meat consumption increases CKD risk, while fruit and vegetables are kidney-protective [72,73]. Plant-contained fats, such as olive oil, have anti-atherogenic effects; in comparison with high protein diets, LPDs, and in particular plant–based LPD, they decrease intraglomerular pressure and have been shown in cohort studies to decrease atherosclerosis risk [73], proteinuria [74] and CKD progression [75,76]. In addition, a vegetable-rich diet is helpful for the prevention, treatment and reversal of the major comorbidities associated with CKD, such as diabetes, hypertension and CVD [77]. Overall, available studies have shown that plant-based diets are also helpful to prevent and treat several CKD complications [78,79]. Adherence to a plant-based diet has been associated with all-cause mortality [78]. In addition, plant-based LPD may protect CKD patients from insulin resistance and inflammation [70], which are common complications of CKD [80,81], since fruits, vegetables and legumes contain high amounts of antioxidants and phytochemicals. Animal proteins are rich in sulfur-containing AAs, such as methionine and cysteine, which increase both the acid load and, in the blood, potentially toxic intermediate products, such as homocysteine and SAH, while plant-based LPDs are less rich of methionine and are naturally alkaline. Acidosis, even mild, causes accelerated muscle catabolism, osteoporosis and, at low bicarbonate levels, hypoalbuminemia [30]. Bases contained in plants, such as citrate, may mitigate metabolic acidosis and slow CKD progression; in addition, a reduction of dietary acid load raises renal uric acid excretion and decreases serum uric acid [82], a putative effector of cardiovascular complications of CKD [83,84]. Moreover, plant phosphates have a lower bioavailability than animal phosphates; therefore, plant-based diets might enable better control of hyperphosphatemia. A recent study on diets containing equivalent amounts of protein, but from different sources, has shown that vegetarian diets administered to non-diabetic CKD patients resulted in reduced plasma phosphate, parathyroid hormone and fibroblast growth factor-23 [85].

Finally, increased fiber intake, associated with plant-based LPDs, might shift the gut microbiota towards reduced production of uremic toxins. A current hypothesis is that a high fiber content in a diet might act as modulator of transcription factors involved in inflammation and oxidative stress [86].

## 10. Plant Proteins Are Less Anabolic than Animal Protein: Short-Term Studies

Plant proteins contribute to a large part of the daily protein intake in the world population [87]; however, plant proteins are less anabolic than animal proteins [13,14]. Essential AA contents of plant proteins such as those contained in oat, lupine and wheat are lower than animal-based proteins (whey, milk, casein and egg) and muscle protein [88]. In particular, two essential AA, methionine and lysine, are lower in plant proteins compared with animal proteins (and muscle protein) [88]. Recently, Schmidt et al. [89] in the Oxford arm of the European Prospective Investigation into Cancer and Nutrition Study, observed that intake of 18 dietary AAs was highest in meat-eaters, followed by fish-eaters, then vegetarians and lowest in vegans (up to 47% lower than in meat-eaters). Plasma concentrations of methionine, tryptophan and tyrosine were highest in fish-eaters and vegetarians, followed by meat-eaters, and lowest in vegans, suggesting dietary deficiency for some AAs in subjects on a vegan diet.

The lower anabolic properties of plant protein sources may also be attributed to a greater splanchnic first-pass and urea production of plant-derived AA compared with animal proteins [14]. In addition, plant proteins are less digestible than animal proteins [14]. This could be both due to the different structure of plant vs. animal proteins and/or due to their content in substances such as phytic acid (found in grains and seeds) and protease inhibitors [90].

Several investigators have studied the effect of consuming plant proteins on muscle protein metabolism in rats, pigs and humans, compared to animal proteins [14,91,92,93]. In some of these studies, protein synthesis has been evaluated at the whole body and/or in skeletal muscle in elderly subjects. The majority of these studies have shown that animal proteins have a greater ability to enhance muscle protein synthesis than plant proteins [94,95,96]. Leucine content appears to be critical for the stimulation of muscle protein synthesis [93]. In addition, in rodents, if leucine is added to a plant protein, muscle protein synthesis rates become insignificantly different from an animal protein [91]. In particular, at lower protein intakes (10% of energy), animal proteins stimulate muscle protein synthesis to a greater degree than plant sources. However, at higher protein doses (30% of energy), both animal and plant proteins (which have reached the amount of leucine and essential AA needed to maximally enhance synthesis) cause similar anabolism [91].

## 11. Long-Term Effects of Consumption of Plant Proteins 

In accordance with studies on the acute effects of ingestion of plant proteins, long-term vegetarianism has been associated to reduced skeletal-muscle mass in older women, compared to an omnivorous diet [91]. If long-term consumption of plant-based meals can maintain muscle mass and function has been addressed by several studies [14,91,92]. In an early study, Campbell et al. [93] observed that, during a 12-week period, a meat containing omnivorous diet providing 1.0 g protein/kg/day combined with resistance training, contributed to a greater gain in lean mass and skeletal muscle mass in older men than a lacto-ovo-vegetarian diet containing 0.78 g protein/kg/day. However, when the plant-based protein diet was increased to 1.1 g protein/kg/d, the increase in muscle mass gain was similar with the two diets, suggesting that when protein intake is high (1.1 g/kg), increases in muscle strength and size are not influenced by the type of protein source in the elderly. Similarly, Wilkinson et al. [94], showed that a plant-based protein diet could enhance muscle mass during a prolonged resistance exercise training, when the amount of plant proteins consumed was at least 30 g/meal. According to these findings, plant proteins need to be provided at sufficient amounts in each meal to substantially increase anabolism.

Increasing the intake of only one plant protein deficient in one of more essential AAs could induce an increased rate of AA deamination, oxidation and ureagenesis. Yang et al. [95] compared changes in muscle protein synthesis following the ingestion of different doses of soy protein and a whey protein isolate in elderly men at rest and after resistance exercise. They observed a direct relationship between protein intake and muscle protein synthesis, which was both dose- and protein source-dependent, with soy isolates showing a reduced effect, as compared to whey protein, to stimulate muscle protein synthesis. These differences were related to the lower leucine postprandial levels and greater enhancement of AA oxidation following the ingestion of soy vs. whey protein. Therefore, it appears important to consume different sources of plant proteins with complementary essential AA composition in order to optimize a full complement of AA intake, especially for the elderly.

Isanejad et al. [96], in a large cross-sectional and prospective study in a cohort of elderly women, observed that a higher total protein intake (1.18 g/kg/day) was directly associated with an increase in lean mass and appendicular lean mass over three years of follow-up as compared to a protein intake of 0.79 g/kg/day. Similar findings were observed both with animal protein and plant proteins (i.e., cereals, vegetables and fruits) intakes. Their data suggest that protein intake may be a modifiable risk factor for sarcopenia in the elderly. In a cross-sectional study including men and women from the Framingham Offspring Cohort in Massachusetts, similar findings were obtained by Sahni et al. [97]. However, Verreijen et al. [98], by using a CT-based measure of cross-sectional muscle area (CSA) in the of the Health, Aging, and Body Composition (Health ABC) Study, came to different results. They explored, in a large (*n* = 1561) prospective cohort study, whether the amount (≥0.8 g/kg vs. <0.8 g/kg and ≥1.2 g/kg vs. <1.2 g/kg) and type (animal or vegetal) of protein intake were associated with 5-year change in mid-thigh muscle CSA in older adults. In their study, no association was observed between energy-adjusted total, animal or plant protein intake and muscle CSA at year 6. This study, which contradicts previous research, suggests that a higher total, animal or vegetable protein intake is not associated with 5-year change in mid-thigh muscle CSA in older adults. In particular, the results of this study are different from those reported by Houston et al. [99], who, using data from the same Health ABC study and DEXA as a measure of lean body mass, observed an association between protein intake and change in lean mass. However, Chan et al. [100] did not find any association between total protein intake and longitudinal change in appendicular lean mass over a 4-year period in a large cohort of elderly subject; in this study, it was underlined that the mean relative total protein intake was 1.3 g/kg, which is higher than the protein requirement recently proposed for older people > 65 years (i.e., 1–1.2 g/kg body weight); the authors argue that this high protein intake was one of the main reasons for finding no effect. More recently, in a study conducted in Japan, elderly people with a higher skeletal muscle index consumed more energy and nutrients and more vegetables than those with a lower skeletal muscle index [101], a finding suggesting that diet is important in preventing muscle loss among the elderly. More recently, Bradlee et al. [102], in data from the Framingham Offspring study, observed that higher intakes of protein-containing foods were associated with a higher percent of muscle mass over nine years, particularly among women. Men and women with higher intakes of foods from animal sources had a higher percent muscle mass regardless of activity; the beneficial effects of plant-based protein foods were only evident in physically active adults. Of note, active subjects with higher intakes of animal or plant protein-source foods had 35% lowest risks of functional decline. Among less active individuals, only those consuming more animal protein-source foods had reduced risks of functional decline (HR: 0.71; 95% CI: 0.50–1.01) [102] These findings suggest that higher intake of animal-protein foods, alone or in combination with physical exercise, is associated with preservation of muscle mass and functional performance in older adults. Similarly, in the Framingham Third Generation Study, Mangano et al. [103] showed that total protein intake was positively associated with appendicular lean mass, but with no difference across protein food clusters (fast food and full-fat dairy, fish, red meat, chicken, low-fat milk and legumes) in middle-aged subjects. Notably, they observed no effect of protein intake on bone density. Very recently, Hruby et al. [104] observed that higher protein intake across adulthood was associated with significantly lower risk of losing functional integrity in aging, an association that was evident only among women. Their results are important and are consistent with the concept that higher protein intake is related to lower risk of frailty.

Taken together, even if with fluctuating results, several long-term studies indicate that a higher protein intake has favorable effects on muscle mass and function in elderly subjects, and that plant protein sources, at low protein intakes (≈0.8 g/kg/day) have lower ability to stimulate protein synthesis in muscle and to cause muscle mass gain as compared to animal proteins. However, the gap between the anabolic effects of plant- and animal-based proteins can be filled with an adequate (1.1–1.2 g/kg) plant protein intake. 

## 12. Are LPD and Plant-Based LPD Safe in Patients with CKD?

In the MDRD study [105], which involved 585 middle aged (mean age 51 years) CKD patients assigned to a usual-protein diet (1.3 g/kg/day), a low-protein diet (0.58 g/kg/day) or a VLPD (0.28 g/kg/day), supplemented with a KA/AA mixture (0.28 g/kg/day), mean values for various indices of nutritional status remained within the normal range during follow-up in each diet group. These analyses suggest that the LPD and VLPDs are nutritionally safe for periods of two to three years. Nonetheless, in the MDRD study, both protein and energy intake declined with time and there were small but significant declines in various indices of nutritional status [105]. Therefore, it is prudent that in subjects who may develop initial signs of protein-energy wasting (PEW) or acute kidney injury (AKI), higher DPI targets are temporarily used until wasting or AKI is resolved [94].

International agencies set a requirement protein intake at 0.66 g/kg body weight/day in adults, whereas the sum of the recommended daily allowance (RDA) of the essential AAs is ∼0.2 g/kg body weight per day. Plant-based LPD may contain up to 40% of animal proteins. As discussed above and based on the recommended RDA for safe DPI ranges, it is highly unlikely that the targeted DPI of 0.8 g/kg/day with 40% animal protein sources (corresponding to 0.32 g/kg/day of animal protein in a 70-kg man) will engender PEW in clinically stable individuals [10,106,107]. However, a chronic use of low quality plant-based LPD at more restricted intakes (i.e., at 0.6 g/kg, containing 40% of animal protein = 0.24 g/kg/day of animal proteins in a 70 kg man) may not meet the requirements and cause wasting in an elderly subject [69].

Hyperkalemia is another potential concern about a vegetable-rich diet [108,109,110]. Until some time ago, vegetables were restricted in CKD patients, owing to the rising concerns of hyperkalemia in subjects on RAAS inhibition therapy. In a recent metanalysis, Morris et al. [108] observed that very-low-quality evidence supports the current opinion that dietary potassium restriction reduces serum potassium and risk of death in CKD; these authors suggested that high-quality randomized controlled trials are needed to demonstrate this hypothesis. The risk of hyperkalemia may be, however, diminished by the use of newly available potassium-binders [109,110].

## 13. How to Increase Anabolism Induced by Plant-Based LPD?

Despite encouraging results from available cross-sectional studies and observational trials, renal nutrition guidelines [111,112] do not recommend the choice of a plant-based LPD in all CKD patients. While waiting for randomized controlled trials, it may be reasonable for clinicians to consider observational data and tailor low protein diets according to patients’ adherence and preferences [48]. It also appears important to plan strategies that may potentiate the anabolism induced by plant-based LPD. Various approaches have been applied to increase the anabolism induced by plant protein [14]. These may include: (1) creating diets containing different plant protein sources to provide a high-quality AA profile; (2) consumption of greater amounts of plant protein; (3) combining plant and animal proteins; (4) supplementing plant-based LPD with essential AA or KA; (5) raising plant sources to obtain high-quality AA profiles.

### 13.1. Creating Diets Containing Different Plant Protein Sources to Provide a High Quality AA Profile

There are large differences in essential AA contents and AA composition between various plant proteins. However, combinations of various plant proteins can provide protein characteristics that reflect the typical pattern of animal proteins [14]. As an example, casein, egg and potato proteins have excellent quality DIAAS scores, and whey and soy proteins are classified as high-quality proteins (average DIAAS ≥ 75). On the contrary, gelatin, rapeseed, lupine, canola, corn, hemp, fava bean, oat, pea and rice proteins are classified in the no quality claim category (DIAAS < 75). Potato, soy and pea proteins can be used in protein mixtures to complement a broad range of plant proteins, leading to higher DIAAS [16]. Legumes (which are deficient in sulfur AAs) are more often combined with cereal proteins (which are lysine-deficient) [113,114] in order to provide patients with a full AA pattern to meet with protein synthesis. In studies on the effect of combining cereals with legumes on protein digestibility [115,116,117] and protein metabolism [114,118,119], better digestibility and protein retention been observed as compared to using individual sources. A limiting AA in these combinations was lysine. This approach has also been successfully used in middle-aged patients with CKD [120]. However, the nutritional benefits of these combinations have not been investigated in older CKD subjects.

### 13.2. Combining Plant and Animal Proteins

It is important to remember that many studies evaluated the anabolic properties of single plant protein sources. However, plant proteins are not commonly eaten alone; they are generally consumed as part of a meal containing various other protein sources. Plant-based LPD may contain up to 40% of animal proteins; therefore, many of these diets (with the exception of the vegan diet) by definition are comprehensive of both animal and plant proteins. The Mediterranean diet, which contains dairy products, fish and poultry in low-to-moderate amounts, is an example of a blended plant and animal protein which has been successfully used in CKD patients [120]. The effect of blending plant and animal proteins for muscle protein synthesis has been evaluated in a few studies in healthy subjects. In young subjects, after resistance exercise, the ingestion of 19 g of a mixture of plant and animal proteins (milk and soy) versus 18 g of whey protein was able to increase to a similar extent both muscle protein synthesis and the activation of the intracellular anabolic signal [121]. Additionally, in older subjects [122], eating 30 g of a mixture of soy and milk proteins after resistance exercise was followed by similar increase in the AA levels observed after eating the same amount of whey protein alone; furthermore, muscle protein synthesis and degradation, and net protein balance, did not differ between the two groups [122]. Therefore, also in elderly subjects, combining plant- with animal-based proteins in sufficient amounts can activate muscle protein anabolism in a similar way to high quality proteins. However, this issue has not been yet addressed in elderly CKD subjects.

### 13.3. Consumption of Greater Amounts of Plant Proteins

Even if plant-based diets may be used less efficiently and be preferentially oxidized, plant- and animal-based diets have been shown to have similar effects on nutrition when protein intake is adequate (>1–1.1 g/kg/day) in subjects with normal renal function. This option may be attractive for CKD patients who are not compliant to an LPD, or undergo some degree of wasting under plant-based LPDs.

### 13.4. Supplementing Plant-Based LPD with Essential AA/KA

It has been well-established that protein ingestion strongly increases muscle protein synthesis rates [123]. Diets containing animal proteins are richer in essential AAs than plant-based diets. Leucine, in particular, has been identified as the key factor stimulating the post-prandial rise in muscle protein synthesis [123]. Supplementation of a plant-based LPD with AA/KA is also attractive because the free essential AAs used to fortify plant-based proteins are digested and absorbed faster than their constitutive AAs, as suggested by Dardevet et al. [124]. Lysine or the sum of methionine and cysteine are the limiting AAs in many plant proteins [14]; however, plasma cysteine pools are higher in CKD patients, while lysine pools are often normal and BCAAs are reduced as compared to heathy subjects. Accordingly, lysine and BCAA/BCKA, but not cysteine, may be offered in supplements to elderly CKD patients on a plant-based LPD, if they develop signs of malnutrition. Supplementation of soy protein with BCAA increased whole-body protein and skeletal muscle, and decreased urea synthesis in healthy elderly and even more in patients with chronic obstructive pulmonary disease [125,126]. Recently, Fuchs et al. [127] observed that ingestion of 6 g BCAA, 6 g branched-chain ketoacids (BCKA) and 30 g milk proteins equally increased myofibrillar protein synthesis in healthy old males; however, the postprandial protein synthesis increase following the ingestion of both BCAA and BCKA was short-lived, while protein synthesis rates remained high following the ingestion of an equivalent amount of intact milk protein. This finding suggests, on the one hand, that in addition to BCAA/BCKA, other essential AAs need to be provided to allow a prolonged postprandial increase in muscle protein synthesis rate; on the other hand, it gives support to the hypothesis that supplementing a plant-based LPD with AA/KA may be strongly anabolic. However, not all studies have clearly shown that leucine supplementation is anabolic in the elderly [128,129]. Recently, two studies using leucine in combination with whey protein and vitamin D showed an increase in appendicular lean mass [130] and fractional protein synthesis rate [131] in older adults. 

AAs and KAs are currently associated to an LPD or a VLPD to meet protein/AA requirements in patients with CKD4-5 [61]; these supplements are likely anabolic, phosphate-free and have phosphate-chelating properties. In addition, the KA and hydroxyacid analogues provide EAA precursors without the N load from EAAs and generate less toxic metabolic products than similar amounts of protein from LPDs. The protein component of the SVLPDs does not need to be of high quality. Aparicio et al. [132] demonstrated that in carefully monitored CKD patients, vegetarian diets with a low or even a very low protein content supplemented with KA provided a satisfactory nutritional status. In an early study, Barsotti et al. [120] observed that transition from a traditional 1.0–1.3 g protein/kg/day diet to a 0.7 g protein/kg/day vegan diet was not associated with substantial changes in serum total protein or albumin in patients with diabetes. It is interesting that these authors combined cereal and legumes, which are complementary for essential AAs. In a recent study by Piccoli et al. [133], in pregnant patients with severe kidney impairment and/or nephrotic proteinuria, a 0.6–0.7 g/kg/day vegetarian diet supplemented with AA/KA turned out to be sufficient for the maintenance of satisfactory nutritional status during the pregnancy and after delivery, even in breast-feeding women. Moreover, most of the children of the vegetarian women had normal intrauterine growth and developed normally between 1 month and 7.5 years from delivery. Recently, Garneata et al. [134] conducted a prospective, randomized and controlled trial of safety and efficacy of a KA-supplemented vegetarian VLPD (sVLPD) compared with a conventional low-protein diet (LPD). After 3 months, compliant patients were randomized to sVLPD (0.3 g/kg vegetable proteins and 1 cps/5 kg KA per day) or continued LPD (0.6 g/kg per day) for 15 months. Adjusted numbers were treated (NNTs; 95% confidence interval) to avoid a composite primary end point in the intention to treat and per-protocol analyses in one patient were 4.4 (4.2 to 5.1) and 4.0 (3.9 to 4.4), respectively, for patients with eGFR < 30 mL/min per 1.73 m^2^; adjusted NNT (95% confidence interval) to avoid dialysis was 22.4 (21.5 to 25.1) for patients with eGFR < 30 mL/min per 1.73 m^2^ but decreased to 2.7 (2.6 to 3.1) for patients with eGFR < 20 mL/min per 1.73 m^2^ in intention to treat analysis. Compliance to diet was good, with no changes in nutritional parameters and no adverse reactions. 

The results of the available studies suggest that a vegetarian diet, when combined with low protein intake and AA/KA supplementation, appears to be suitable for CKD patients and to provide adequate nutrition in patients who are compliant to this treatment; however, the efficacy of such dietary strategies on postprandial muscle protein synthesis remains to be studied. Future research comparing the anabolic properties of a variety of plant-based proteins could define the preferred protein sources to be used in nutritional interventions to support skeletal muscle mass gain in patients with CKD.

### 13.5. Genetic Engineering and Other Approaches to Improve the Quality of Plant Proteins 

Genetic engineering to enhance the essential AA profile of plant proteins may become a new strategy to improve muscle protein synthesis after plant protein feeding. An example is given by quality protein maize (QPM), a modified maize obtained by selectively breeding maize with a mutation of a gene named opaque-2, which leads to increased lysine and tryptophan contents [14,135]. QPM has a nearly two-fold higher lysine content compared to conventional maize close to the WHO/FAO/UNU-recommended lysine requirements for adult humans (45 mg/g protein). In young children [136], as compared to conventional maize, the growth rate was increased by 15% by the administration of QPM for one year. However, research is needed to evaluate the impact of eating plant proteins with improved AA composition on the postprandial muscle protein synthesis and enhancement in muscle mass in the long-term.

Another possible approach targets the intestinal microbiota by adding β-glucans to plant proteins. The enrichment of whole-grain barley and oat flours with β-glucans [137], which are soluble dietary fibers, can actively lower/reduce blood LDL-cholesterol and decrease insulin resistance, an effect which may improve protein metabolism [80].

## 14. Conclusions

Current evidence shows that high-quality protein profile of dietary proteins enhances the efficiency of muscle protein metabolism, especially in the elderly. The LPD was historically used to delay uremic symptoms and decrease N derived catabolic products in patients with CKD. In recent years, it has become evident that nutritional intervention is a necessary approach to prevent wasting and reduce CKD complications and disease progression. In the short-term, clinically stable CKD patients can efficiently adapt their muscle protein metabolism to an LPD containing 0.55–0.6 g protein/kg, containing high-quality protein, or to an AA/KA supplemented VLPD. Plant-based protein sources that are rich in fiber, alkali and micronutrients have many favorable effects, but they are potentially less anabolic than animal-based proteins at low DPI (<0.78 g/kg/day) and the muscle response to plant-based LPD in CKD is not known. Possible approaches to increase the anabolic potential of plant-based LPD include supplementation with essential AA/KA, improving protein quality, mixing different plant proteins and finally blend plant- with animal-based protein sources. Utilizing cereal and legume composite mixes could help to improve nutritional properties. These promising strategies have been studied in younger or middle-aged individuals, but still need to be examined in elderly CKD subjects. 

## Figures and Tables

**Figure 1 nutrients-13-00083-f001:**
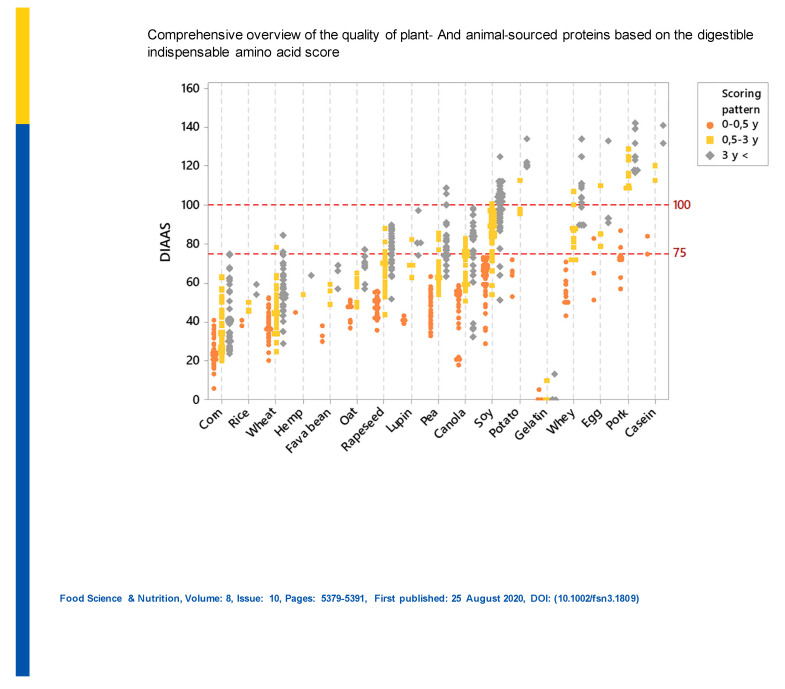
Overview of the quality of individual proteins based on their digestible indispensable amino acid score (DIAAS) [16].

**Figure 2 nutrients-13-00083-f002:**
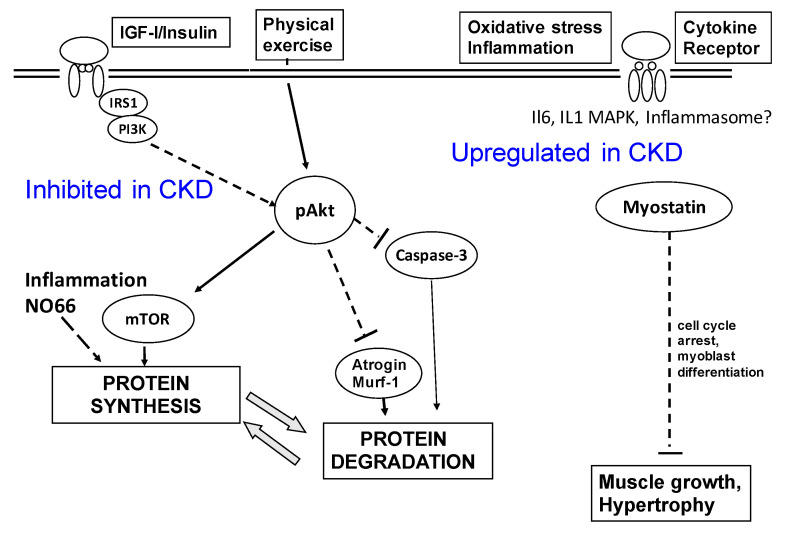
Scheme of mechanisms by which chronic kidney disease (CKD) impairs muscle protein metabolism. The inhibition of PI3-kinase/Akt pathway is associated with an inhibition of IGF-I and insulin actions on muscle protein synthesis and growth; in addition, downregulation of p-Akt increases protein degradation, by acting on atrogin-1, MuRF-1 and caspase-3. Overexpression of Myostatin by inflammation results in cachexia by increased protein degradation because of upregulation of atrogin-1 and MuRF-1. Inflammation upregulates nucleolar protein 66 (NO66), which suppresses both ribosomal DNA transcription and muscle protein synthesis via a demethylase mechanism.

## Data Availability

No new data were created or analyzed in this study. Data sharing is not applicable to this article.

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
