# Peer review of "Low Protein Diets and Plant-Based Low Protein Diets: Do They Meet Protein Requirements of Patients with Chronic Kidney Disease?"

_nutrients, 2020, doi:10.3390/nu13010083_

Round 1

Reviewer 1 Report

This review systematically demonstrated the benefits of a low-protein diet in CKD patients in the context of adequate nutrition to maintain nitrogen balance and prevent muscle catabolism.

They presented plant protein as a viable option and showed the added value of fiber, alkalinity, and trace element supplementation.

They also described the importance of supplementing with KA/AA, combining multiple plant proteins, and consuming a combination of animal proteins to maximize the effectiveness of plant proteins.

I consider this review was well-written and the content was also reasonable.

Line 120 DIIAS → DIAAS

Line 498 protein-energy wasting or → protein-energy wasting (PEW) or

Author Response

Reviewer 1

This review systematically demonstrated the benefits of a low-protein diet in CKD patients in the context of adequate nutrition to maintain nitrogen balance and prevent muscle catabolism.

They presented plant protein as a viable option and showed the added value of fiber, alkalinity, and trace element supplementation.

They also described the importance of supplementing with KA/AA, combining multiple plant proteins, and consuming a combination of animal proteins to maximize the effectiveness of plant proteins.

I consider this review was well-written and the content was also reasonable.

Answer:We thank the Reviewer for constructive comments. The revised version of the MS contains the suggested changes. In addition the MS has been reviewed as for English structure and grammar. We hope that with the changes may, the MS may be acceptable.

Question: Line 120 DIIAS → DIAAS

Answer: This acronym has been corrected.

Question: Line 498 protein-energy wasting or → protein-energy wasting (PEW) or

Answer: The term: PEW has been explained.

Reviewer 2 Report

Dear authors, the present review deals with a very popular topic of nowadays,  the use and safety of plant-based low protein diets in CKD patients.

The manuscript is well organized and constructed, giving the reader the up today relative information, the missing points and the areas that need further research. The topic is of great interest in renal dietitians, nephrologists and the renal patients themselves of course.

However, the text needs a thorough reading so as to correct plenty of typing mistakes. For example, please correct the affiliation numbers in line 8 and 9 (11 and 33 instead of the correct 1 and 3 respectively).

The fact that "histidine has been added to the list of essential AA", should been mentioned before in lines 31-32 instead of the current one (75).

Line 187, please write the full name before using the abbreviation BCAA 

Line 211 and 233, please write the full name before using the abbreviation GH and Cbl respectively. 

Line 235, "A greater benefit for CVD was observed among participants with lower plasma". I suppose you mean "higher

line 299, a reference is missing

Author Response

Reviewer 2

Dear authors, the present review deals with a very popular topic of nowadays,  the use and safety of plant-based low protein diets in CKD patients.

The manuscript is well organized and constructed, giving the reader the up today relative information, the missing points and the areas that need further research. The topic is of great interest in renal dietitians, nephrologists and the renal patients themselves of course.

Question: However, the text needs a thorough reading so as to correct plenty of typing mistakes. For example, please correct the affiliation numbers in line 8 and 9 (11 and 33 instead of the correct 1 and 3 respectively).

Answer: We thank the Reviewer for constructive comments and criticism. The revised version of the MS contains the suggested changes. In addition the MS has been reviewed as for English structure and grammar. We hope that with the changes may, the MS may be acceptable.

Question¸The fact that "histidine has been added to the list of essential AA", should been mentioned before in lines 31-32 instead of the current one (75).

Answer: The sentence: Laterly (1), histidine was added to the list of essential AA has been anticipated to line 33.

Question:Line 187, please write the full name before using the abbreviation BCAA 

Answer:Thank you. In Line 187 (now line 189) the new sentence starts with: The metabolism of Branched-chain AA (BCAA)..

Question:Line 211 and 233, please write the full name before using the abbreviation GH and Cbl respectively.

Answer:Thank you. We inserted the full name in association with abbreviations. The abbreviation for cobalamin was removed.

Question:Line 235, "A greater benefit for CVD was observed among participants with lower plasma". I suppose you mean "higher

Answer: We apologize for not having clarified the concept. A greater benefit of folic acid supplementation for CVD was observed among participants with lower plasma folate levels and in studies with larger decreases in homocysteine levels. Accordingly, the sentence has been modified as follows:

Line 238: A recent meta-analysis [43]. indicated a 10% lower risk of stroke and a 4% lower risk of overall CVD with folic acid supplementation. A greater benefit of folic acid supplementation for CVD was observed among participants with lower plasma folate levels and in studies with larger decreases in homocysteine levels.

Question:line 299, a reference is missing

Answer: Thank you. Quotation  of Ref 47 has been introduced.